REGISTERED REPORT PROTOCOL

# Effectiveness of aerobic physical exercise on depression symptoms in adults: A protocol for developing a systematic review and meta-analysis of randomized clinical trials

Larissa Nayara de Souza[1]*, Silvana Medeiros de Araújo[1], Eva da Silva Paiva[1], Alícia Eliege da Silva[1], Joel Florêncio da Costa Neto[1], Juvêncio César Lima de Assis[1], Isis Kelly dos Santos[1], Themis Cristina Mesquita Soares[2], Edson Fonseca Pinto[3], Roque Ribeiro da Silva Júnior[4], Maria Irany Knackfuss[1]*

1 Programa de Pós-Graduação em Saúde e Sociedade pela Universidade do Estado do Rio Grande do Norte, Mossoró, Brazil, 2 Programa de Mestrado Acadêmico em Planejamento e Dinâmicas Territoriais no Semiárido PLANDITES-CAPF-UERN, Pau dos Ferros, Brazil, 3 Programa de Pós-Graduação em Saúde Coletiva PPGScol pela Universidade Federal do Rio Grande do Norte, Natal, Brazil, 4 Programa Multicêntrico de Pós-graduação em Ciências Fisiológicas pela Universidade do Estado do Rio Grande Norte, Mossoro, Brazil.

* mariaknackfuss@uern.br (MIK); larissanay15@gmail.com (LNdS)

This is a Registered Report and may have an associated publication; please check the article page on the journal site for any related articles.

## Abstract

### Background

Depression is a chronic condition that affects millions of people and requires effective interventions. Studies suggest that aerobic exercise can improve mental health and reduce depression symptoms, despite variations in exercise type and intensity.

### Objective

This article presents a protocol for conducting a systematic review with meta-analysis, aimed at investigating the effects of aerobic exercise on reducing depression symptoms in adults.

### Methods

This is a systematic review protocol, which will follow the PRISMA-P 2020 guidelines and has been registered in PROSPERO (no. CRD42024592700). The study search will be conducted in six databases: BVS, Cochrane library, Embase, Epistemonikos, PubMed and SPORTDiscus, using MeSH-based descriptors. Studies will be selected independently by two researchers using the Rayyan 'QCRI' software. Data extraction will be performed using specific forms, while the methodological quality of studies on physical exercise will be assessed with the TESTEX tool, and the risk of bias will be evaluated using the Cochrane RoB 2.0 method. The certainty of evidence will be

**Data availability statement:** Relevant data will be made available with the systemic review publication.

**Funding:** This work was supported by the National Council for Scientific and Technological Development (CNPq), Brazil. The funders had no role in the study design, data collection and analysis, decision to publish, or preparation of the manuscript.

**Competing interests:** The authors have declared that there are no conflicting interests

evaluated through the Grading of Recommendations Assessment, Development, and Evaluation (GRADE).

## Conclusions

It is expected that the systematic review following this protocol will identify the effective dose-response to reduce depression levels and provide an understanding of the mechanisms through which aerobic exercise influences depression.

## Introduction

According to the World Health Organization (WHO), more than 300 million people of all ages suffer from depression, also known as major depressive disorder (MDD) [1]. This situation highlights the urgency of finding effective strategies to address this serious public health issue.

Since it is a condition often aggravated by other diseases, such as substance use disorders, diabetes, and heart diseases, which not only increase the risk of developing depression but are also affected by it, establishing a cycle that further compromises the health and quality of life of those affected [2].

In general, mental health problems, neurological disorders, and substance use disorders account for 13% of the global burden of disease, with depression alone responsible for 4.3% of this total, women are the most affected [3]. In 2019, there were 290.2 million cases of depression worldwide, an increase of 59.28% compared to 1990 [3].

Projections based on the Bayesian age-period-cohort (BAPC) model indicate that, by 2030, global age-standardized rates are expected to remain stable. However, the absolute number of depression cases is likely to increase, potentially reaching 108.9 million among men and 164.9 million among women [4].

The research conducted by Liu et al. [5] reinforces these projections by indicating stability in age-specific incidence rates between 1990 and 2017, along with an increase in the total number of global depression cases, reaching 258 million in 2017, a growth of 49.86% compared to 1990. Regions with low Socio-Demographic Index (SDI) face the greatest impacts, mainly due to the lack of investment in mental health. In these areas, depression often remains undiagnosed or inadequately treated, resulting in higher morbidity and disability.

In the United States, the annual prevalence of depression among adults reached 8.9 million people undergoing pharmacological treatment, with approximately 30.9% (2.8 million) being treatment-resistant [6].

Treatment-resistant depression is defined as the failure of at least two adequate trials with different antidepressants, over a minimum period of four to six weeks, at the maximum recommended dose [7,8].

Given the multifaceted nature of the disorder, significant gaps are still observed in the studies, attributed in part to the diversity of symptoms and the limitations of current technologies to investigate the human brain in real time, both at the circuit

and synapse levels [9]. The condition can be triggered by biological changes [10], as well as by psychosocial influences [11,12].

In view of the aforementioned issues, researchers and scholars in the field have been seeking effective strategies capable of responding, in an individualized manner, to the complexities of depressive disorder, considering its multifactorial nature. In this context, physical exercise has emerged as a promising intervention for mental health, especially among individuals with antidepressant-resistant depression. Furthermore, evidence indicates that even in these cases, physical exercise, when used as an adjunct to pharmacological treatment, demonstrates significant potential for reducing depressive symptoms, including non-remissive cases, as well as improving quality of life, sleep, and vitality [13].

The focus of this review will be the study of exercises with an aerobic predominance, commonly known as 'aerobic exercise,' which are characterized by the primary use of the oxidative pathway for energy production. This type of exercise involves long-duration activities with varying intensity levels, ranging from low (walking), moderate (light jogging, cycling), to high (continuous running near $VO_2max$, competitive cycling, endurance swimming). The energy demand in these efforts is primarily sustained by mitochondrial metabolism [14].

However, the analysis of these exercises should not be based solely on perceived effort or heart rate, as all energy pathways operate simultaneously, varying according to the duration and intensity of the exercise [14]. Thus, efforts are categorized as follows: explosive efforts (up to 6 seconds), dominated by the phosphagen system (ATP-CP); high-intensity efforts (6 seconds to 1 minute), where anaerobic glycolysis predominates but with an increasing aerobic contribution; and prolonged endurance efforts (above 1–2 minutes), where oxidative phosphorylation plays a primary role, although anaerobic glycolysis still contributes during intensity transition phases [14].

Although the benefits of aerobic exercise in improving depressive symptoms are well documented, the existing literature shows high variability in terms of modality, intensity, frequency, and duration of exercise. Furthermore, the underlying mechanisms of these effects are still not fully understood [15].

Therefore, this study is justified by the need to understanding of how different intensities and types of aerobic exercise can influence the reduction of depression levels in adults. Various international guidelines, such as those from the United Kingdom and Australia, recommend physical exercise as part of depression treatment [16,17]. However, these guidelines do not provide clear and consistent recommendations regarding the dose or modality of exercise [18]. In turn, the American Psychiatric Association recommends any dose of aerobic exercise or strength training [19], while the Australian guidelines already suggest a combination of vigorous aerobic and strength exercises at least twice a week [17].

Recent reviews, including network meta-analyses, suggest that both the dose and the modality of exercise may influence clinical outcomes [20]. However, many existing systematic reviews and meta-analyses have methodological limitations that prevent definitive conclusions. In light of this, the isolated investigation of aerobic exercise using a robust and clearly defined methodology may contribute to a more accurate understanding of its effectiveness on depressive symptoms.

A recently conducted systematic review investigating the effect of physical exercise on depression revealed that dance exercises combined with selective serotonin reuptake inhibitors (SSRIs), walking, or running were the treatments most likely to be effective in treating depression. Furthermore, the results indicate that the effect of exercise is proportional to its intensity, with vigorous activities showing better responses. However, the findings did not allow for the precise establishment of a dose-response relationship due to variability in the study protocols analyzed, particularly regarding frequency, duration, and weekly volume of activity [18]. In this regard, the development of this protocol focused exclusively on aerobic exercises represents a viable strategy to guide a future systematic review and refine the results obtained in the scientific literature.

This protocol proposal aims to guide a future systematic review that will seek to answer the following question: what are the effects of aerobic exercise on reducing depression levels in adults? With this protocol, we intend to provide researchers and healthcare professionals with a rigorous and transparent methodological foundation for conducting a systematic review and meta-analysis of randomized clinical trials on the topic.

 

## Methods

The study is a protocol for a systematic review, which will follow the guidelines developed by Page et al. [21] known as the Preferred Reporting Items for Systematic Reviews and Meta-Analyses (PRISMA-P) for systematic reviews and has been registered in the International Prospective Register of Systematic Reviews - PROSPERO, number: CRD42024592700.

### Research question

Thus, the clinical PICOS question, widely recommended to guide systematic reviews [22], will be formulated as follows: Population (adults with depression); Intervention (aerobic exercise); Comparator (other types of exercise/placebo/control); Outcomes (improvement of depressive symptoms); Study types (randomized controlled trials – RCTs).

### Search strategy

The search will be conducted in the following databases: BVS, Cochrane library, Embase, Epistemonikos, PubMed and SPORTDiscus. Additionally, Clinical Trials, Google Scholar (*advanced*), and the references of included studies will be scanned to identify potential studies.

The search strategy will be highly sensitive and systematic, using the official descriptors from the Medical Subject Headings (MeSH) and EMTREE databases, adapted for each database. There will be no restrictions on publication date or language. Descriptors will be combined using the Boolean operators 'AND/OR'. It is worth noting that the descriptors and their synonyms were organized into sets and subdivided into lines, as shown in Table 1.

### Eligibility criteria

We will follow these criterias: Participants will include adults (>18 years) diagnosed with depression, excluding individuals with chronic degenerative diseases or other associated severe psychiatric disorders. The intervention will focus on aerobic exercise (e.g., walking, running), with comparators including other interventions or control groups. The primary outcome will be depressive symptoms, while training volume and frequency will be considered secondary outcomes. Only studies utilizing validated depression assessment scales will be included, such as the Beck Depression Inventory (BDI), Hamilton Depression Rating Scale (HAM-D), Patient Health Questionnaire (PHQ-9), Center for Epidemiologic Studies Depression Scale (CES-D), Montgomery-Åsberg Depression Rating Scale (MADRS), Zung Self-Rating Depression Scale (Zung SDS), Geriatric Depression Scale (GDS), and Hospital Anxiety and Depression Scale (HADS), among others. Eligible study types will be randomized clinical trials (RCTs), no restriction of language, while ongoing clinical trials with preliminary results will be excluded. Additionally, studies involving populations with non-communicable chronic diseases (e.g., hypertension, diabetes), case reports, preprints, narrative and systematic reviews, and observational studies will not be considered for inclusion.

After conducting a high-sensitivity search in the databases, articles will be selected using Rayyan software, a tool specialized in screening studies for systematic reviews and meta-analyses [23]. Two reviewers (L.N.S; M.I.K) will initially be registered on the platform, where they will receive the articles identified by the search. The technical assessment will be divided into two phases: both reviewers will examine the articles in the first phase to determine their inclusion in the review. A third senior and expert reviewer (I.K.S) will review any articles in case of disagreement, and make the final decision on its inclusion.

**Table 1. High-sensitivity search strategy.**

| |
|---|
| #1 "Depression" [Mesh] OR (Depressive Symptoms) OR (Depressive Symptom) OR (Symptom, Depressive) |
| #2 "Exercise" [Mesh] OR (Exercise, Aerobic) OR (Aerobic Exercise) OR (Aerobic Exercises) OR (Exercises, Aerobic) |
| #3 "Randomized Controlled Trials as Topic" [Mesh] OR (Clinical Trials, Randomized) OR (Trials, Randomized Clinical) OR (Controlled Clinical Trials, Randomized) |

After applying the eligibility criteria, the selection process will be conducted in two stages. The titles and abstracts of the articles will be read in the software in the first stage, which must be accepted by at least one of the reviewers for it to proceed to the subsequent phases. Next, the articles will be read in full in the second stage, and acceptance by both reviewers will be required for them to be included in the study. The flowchart will be developed according to the procedure and guidelines established by PRISMA, as shown in Fig 1.

## Data extraction

The following data for each study included in the systematic review will be extracted in detail: first author, year of publication of study; sample characteristics (size, inclusion and exclusion criteria, as well as other relevant participant characteristics, such as physical condition); and methodological details (study type, description of the intervention and control

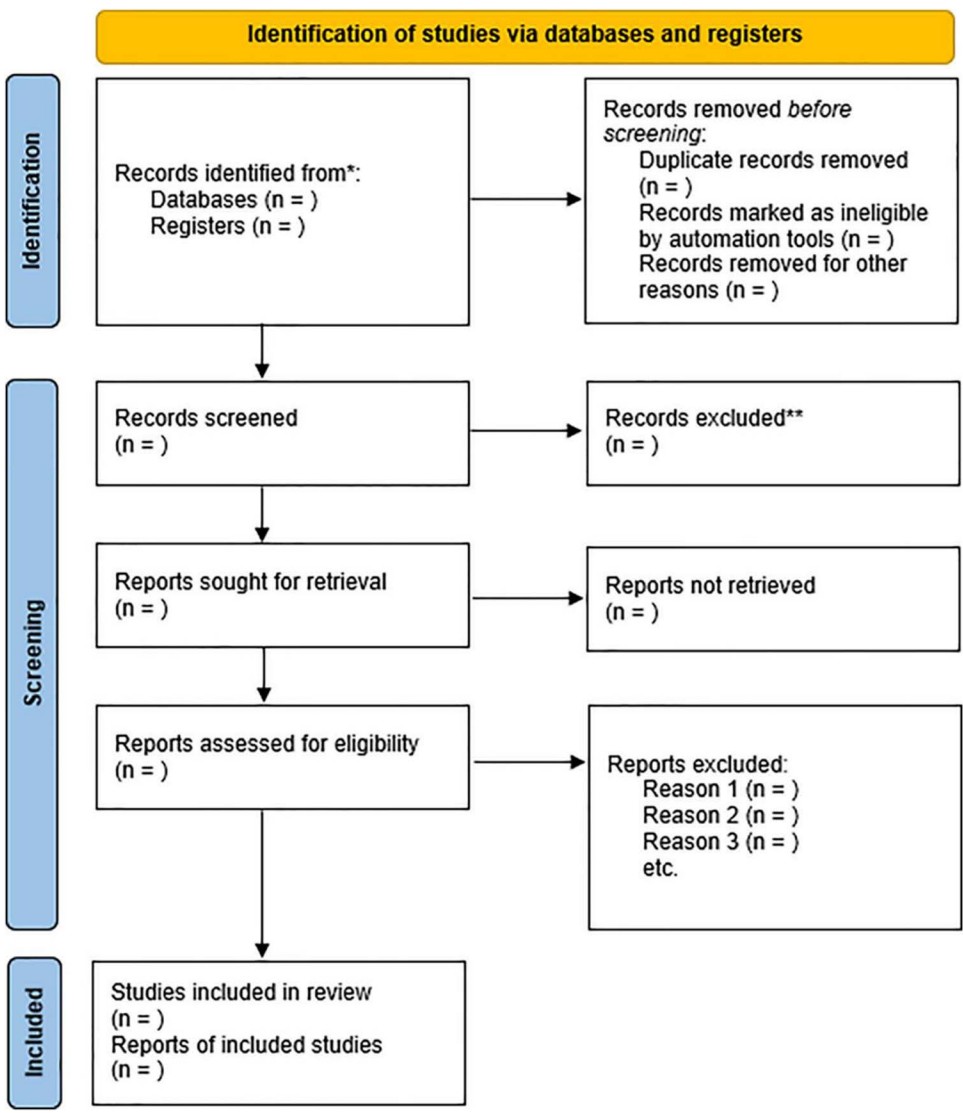

**Fig 1. PRISMA flow diagram for systematic review and meta-analysis.**

groups, with an emphasis on the duration, intensity, and frequency of the exercises, as well as the scales used to measure the outcomes) and main results.

Primary and secondary outcomes will be analyzed focusing on the following aspects: changes in depressive symptoms as primary outcomes; training volume and activity frequency, dropouts and adverse events as secondary outcomes. The validated instruments used to assess these outcomes will also be recorded.

Quantitative results will include effect size, measures of variability such as confidence intervals, standard deviation, and p-values, as well as other data relevant for interpreting the results.

After collecting the described data, a synthesis table will be prepared to group the extracted information, including year of publication, authors, methods, interventions, and main results of each study. This table will facilitate a comparative analysis of the included studies and provide a clearer view of the reported outcomes.

Data for studies reporting multiple outcomes or using different measures for the same outcome will be extracted separately and treated as independent entries, thereby avoiding duplication of results. When more than one scale is used to assess depressive symptoms, the most widely used scale (i.e., the Hamilton Scale) will be prioritized, with a clear justification for the choice.

## Risk of bias assessment

The methodological quality assessment of the included studies will be carried out by two independent reviewers (L.N.S; M.I.K) using the TESTEX tool (Tool for the assEssment of Study qualiTy and reporting in EXercise), which was specifically developed for clinical trials involving physical exercise interventions. This scale, validated by Smart et al. [24], includes criteria addressing both methodological quality and reporting quality of the studies, with a total possible score of 15 points—5 for study quality and 10 for reporting quality. TESTEX stands out for addressing relevant and specific aspects of exercise interventions, such as adherence to the protocol, exercise intensity, and volume, making it more appropriate than other tools for use in exercise-related reviews.

In addition, the Cochrane Risk of Bias 2.0 tool (RoB 2.0) will be used to assess the risk of bias in the included studies [25]. This tool evaluates the following domains: the randomization process, deviations from the intended intervention, missing outcome data, measurement of outcomes, selection of the reported results, and the overall judgment of the risk of bias.

## Certainty of the evidence

The Grading of Recommendations Assessment, Development, and Evaluation (GRADE) is a widely used system for the methodological assessment of scientific studies, aimed at determining the level of evidence and strength of recommendations. This system is based on rigorous criteria to ensure the quality and reliability of the presented evidence. In this protocol, the GRADE approach will be applied to assess the certainty of the evidence for each outcome included in the systematic review and meta-analysis. The evaluation will be based on the following domains: risk of bias, inconsistency of results, indirectness of evidence, imprecision of estimates, and potential publication bias. Two reviewers will independently assess each domain, and disagreements will be resolved by consensus or by a third reviewer. The results will be summarized in a GRADE evidence profile and presented in a Summary of Findings table. GRADE categorizes both the level of evidence and the strength of recommendations into four categories: high, moderate, low, and very low [26].

## Data synthesis

The mean and standard deviation (SD) values reflecting the change in depression scores from baseline to post-intervention will be extracted from each study for pooling effects. SD will be calculated using a previously described

formula for studies reporting standard error (SE) or 95% confidence intervals (CIs). The effect models (fixed or random) will be used to estimate the treatment effect (aerobic exercise) for depression in adults. The choice of model for analysis will depend on the similarity of the effects of interest among the studies: fixed-effect models will be applied to similar studies, while random-effects models will be used for the others.

The meta-analysis will be conducted using fixed-effect or random-effects models, depending on the heterogeneity among the included studies. The degree of heterogeneity will be assessed using the I² statistic. A fixed-effect model will be applied when heterogeneity is low (I² ≤ 50%), while a random-effects model will be used when heterogeneity is moderate or high (I² > 50%) [27].

Heterogeneity among studies will be assessed using the I² index, which quantifies inconsistency among study results, and Cochran's Q test, which evaluates the statistical significance of heterogeneity. Heterogeneity will be interpreted according to I² values following the recommendations of the Cochrane Handbook for Systematic Reviews of Interventions. Statistical significance will be set at P < 0.05. RevMan version 5.3.5 will be used for all analyses, including meta-analysis if feasible [28]. One review author will be responsible for entering data into the software, while a second author will check data accuracy.

The presence of bias in relation to estimates against sample size will be analyzed using a funnel plot, which will be generated only if at least 10 studies are included in the meta-analysis, as recommended to ensure reliability of asymmetry assessments [27].

For the assessment of publication bias, Egger's statistical test will be implemented, a sophisticated tool used to enhance the consistency of funnel plot analysis [29]. In cases where publication bias is identified, a technique that may be used to correct it is the trim-and-fill method, which adjusts the distribution of studies in an asymmetric funnel plot [30].

We will analyze a forest plot by considering the effect size and 95% confidence intervals (CIs), heterogeneity (I²), study weight, and the pooled effect size. All of these points should be interpreted in relation to the null value, and potential publication bias should be evaluated using a funnel plot or Egger's test.

## Subgroup analyses

Subgroup analyses will be conducted to evaluate the effectiveness of aerobic exercise in different contexts:

• Exercise intensity: classification based on perceived exertion rate or heart rate (e.g., light, moderate, vigorous).

• Intervention duration: comparison between short and long-duration interventions.

• Type of exercise: differentiation among aerobic exercise modalities (walking, running, cycling, etc.).

• Participant characteristics: analyses based on age groups (i.e., adults >18 years), gender, and mental health status.

Subgroup analysis is essential for exploring the effectiveness of aerobic exercise in different contexts and populations. Different exercise intensities (light, moderate, vigorous) may have variable effects on depression, and this differentiation will help identify the most effective modalities. Intervention duration can significantly influence outcomes, making it important to compare short- and long-duration interventions. Participant characteristics, such as age and gender, may also moderate the effects, justifying separate analysis for these groups.

## Sensitivity analysis

A sensitivity analysis will be conducted to assess the robustness of the results in relation to the risk of bias in the included studies. Studies with a high risk of bias will be identified, considering the worst judgment in any of the assessed domains, excluding blinding domains due to the nature of physical exercise interventions. Excluding these studies will allow us to evaluate whether they significantly influence the overall results of the meta-analysis.

### Strategy for handling missing data

Data imputation methods or intention-to-treat analysis will be applied to handle missing data, ensuring that the meta-analysis results are complete and representative.

### Publication bias

Publication bias will be examined through funnel plots and Egger's test. If significant bias is detected, correction techniques, such as the trim-and-fill method, will be considered to adjust the results.

### Presentation of results

The results will be presented in forest plot graphs, which will illustrate the combined effects of the interventions and the heterogeneity among the studies. Additionally, a narrative synthesis will discuss the clinical implications of the findings, the limitations of the included studies, and the research gaps that need to be addressed in future investigations.

## Discussion

The literature extensively documents the benefits provided by aerobic exercise for the clinical improvement of adults with depression. However, the topic still requires further exploration due to the complexity of the variables involved in psychological responses. Aerobic exercise has been shown to be a potential strategy for treating depression in different populations, being comparable to various therapeutic approaches, including medication, psychotherapy, and other exercise-based therapies. Studies aim to evaluate this relationship to enhance clinical interventions for treating this condition. For instance, in the research conducted by Olson et al. [31], moderate-intensity aerobic exercise not only helped reduce depressive symptoms but also contributed to cognitive control, suggesting benefits beyond emotional relief. Similarly, a study focused on geriatric interventions demonstrated that short-duration aerobic exercise produced promising results in hospitalized patients, indicating that even brief interventions may be effective [32].

An analysis of exercise intensity in studies indicated that although more intense activities may offer additional benefits, the potential gains for reducing depressive symptoms are not limited to high-intensity levels. Recent research has shown that both light and intense exercises promoted relief from depressive symptoms, with no significant differences between groups of varying intensities [33,34]. This suggests that exercise intensity can be adapted according to individual needs for the initial treatment of depression.

Given the gaps in the literature that have yet to establish an optimal dose-response of physical exercise in the management of depression, a clinical trial evaluating the impact of different intensities (light, moderate, and vigorous) on depressive symptoms showed a significant reduction in symptoms across all groups, with no statistically relevant differences between intensities. These findings highlight the need for comprehensive systematic reviews, such as the present one, to identify the most effective combination of exercise intensity and volume, considering the specificities of different clinical profiles [33].

In summary, the benefits of exercise go beyond improvements in mental health parameters; research reveals connections with cognitive and physiological aspects. A study comparing aerobic exercise with stretching, for instance, showed improvements in visuospatial memory and cardiorespiratory capacity in individuals who engaged in aerobic exercise [35]. These results paradoxically emphasize what existing literature suggests: physical activity can promote neurobiological changes, such as increased neuroplasticity and enhanced cognitive function.

Given the challenges of adherence to exercise programs among individuals with depressive characteristics, the study "DEMO-II," for example, found difficulties in maintaining high levels of participation, which may have limited the results [35]. Consequently, the research suggested strategies that consider patients' motivation and intrinsic preferences. Low-intensity, less strenuous exercises may be more appealing to increase adherence and ensure continuity of treatment [35].

Research evaluating the efficacy of exercise as an addition to standard treatment revealed that it can enhance the effects of traditional interventions, promoting improvements in quality of life and sustained reductions in depressive symptoms. Although exercise is not yet considered an essential component on its own, the findings suggest it has great potential to be integrated as a significant part of therapeutic strategies, especially for patients who do not fully respond to conventional treatments [36].

The systematic review studies conducted so far, although following rigorous methodologies and providing relevant contributions to the scientific field, present some methodological limitations. Anticipated limitations and challenges include the following issues: the use of small sample sizes, lack of clarity regarding the specific mechanisms through which exercise contributes to the improvement of depression, and variations in the intensity protocols adopted. In addition, one of the main methodological challenges expected in the conduction of this review is the high heterogeneity among the studies to be included. Differences may arise in exercise modalities, levels of depressive symptoms, different intensities, frequency and duration of exercise, presence of other comorbidities, supervised versus unsupervised exercise, individual versus group exercise, and gender differences, as men and women may respond differently. However, this review aims precisely to highlight these issues and deepen the understanding of their implications.

## Conclusion

The systematic review of RCTs aims to present findings on reducing depression levels in adults and to identify the most effective dose-response for this reduction, specifying which intensities, frequencies, and durations of aerobic exercise produce the most favorable outcomes. Additionally, the review is expected to offer in-depth understanding of the biological and psychological mechanisms through which aerobic exercise influences depression levels.

## Author contributions

**Conceptualization:** Larissa Nayara de Souza, Silvana Medeiros de Araújo, Eva da Silva Paiva, Alícia Eliege da Silva, Joel Florêncio da Costa Neto, Juvêncio César Lima de Assis, Isis Kelly dos Santos, Themis Cristina Mesquita Soares, Edson Fonseca Pinto, Roque Ribeiro da Silva Júnior, Maria Irany Knackfuss.

**Data curation:** Larissa Nayara de Souza, Silvana Medeiros de Araújo, Eva da Silva Paiva, Alícia Eliege da Silva, Joel Florêncio da Costa Neto, Juvêncio César Lima de Assis, Maria Irany Knackfuss.

**Formal analysis:** Isis Kelly dos Santos, Themis Cristina Mesquita Soares, Edson Fonseca Pinto, Roque Ribeiro da Silva Júnior, Maria Irany Knackfuss.

**Funding acquisition:** Maria Irany Knackfuss.

**Investigation:** Larissa Nayara de Souza, Silvana Medeiros de Araújo, Eva da Silva Paiva, Alícia Eliege da Silva, Joel Florêncio da Costa Neto, Juvêncio César Lima de Assis, Maria Irany Knackfuss.

**Methodology:** Larissa Nayara de Souza, Isis Kelly dos Santos, Themis Cristina Mesquita Soares, Edson Fonseca Pinto, Roque Ribeiro da Silva Júnior, Maria Irany Knackfuss.

**Project administration:** Larissa Nayara de Souza, Isis Kelly dos Santos, Maria Irany Knackfuss.

**Supervision:** Isis Kelly dos Santos, Maria Irany Knackfuss.

**Validation:** Larissa Nayara de Souza, Isis Kelly dos Santos, Themis Cristina Mesquita Soares, Edson Fonseca Pinto, Roque Ribeiro da Silva Júnior, Maria Irany Knackfuss.

**Visualization:** Larissa Nayara de Souza, Maria Irany Knackfuss.

**Writing – original draft:** Larissa Nayara de Souza, Silvana Medeiros de Araújo, Eva da Silva Paiva, Alícia Eliege da Silva, Joel Florêncio da Costa Neto, Juvêncio César Lima de Assis, Maria Irany Knackfuss.

**Writing – review & editing:** Larissa Nayara de Souza, Isis Kelly dos Santos, Themis Cristina Mesquita Soares, Edson Fonseca Pinto, Roque Ribeiro da Silva Júnior, Maria Irany Knackfuss.

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
