## [Decision Letter · Decision Letter 0]

26 Jan 2025

PONE-D-24-51714Effectiveness of aerobic physical exercise on depression symptoms in adults: A protocol for a systematic review and meta-analysis of randomized clinical trialsPLOS ONE

Dear Dr.  Souza,

Thank you for submitting your manuscript to PLOS ONE. After careful consideration, we feel that it has merit but does not fully meet PLOS ONE’s publication criteria as it currently stands. Therefore, we invite you to submit a revised version of the manuscript that addresses the points raised during the review process.

**ACADEMIC EDITOR: Dear Author, please make changes based on the comments provided by the reviewers.**

We look forward to receiving your revised manuscript.

Kind regards,

Zulkarnain Jaafar

Academic Editor

PLOS ONE

Journal Requirements:

2. In your cover letter, please confirm that the research you have described in your manuscript, including participant recruitment, data collection, modification, or processing, has not started and will not start until after your paper has been accepted to the journal (assuming data need to be collected or participants recruited specifically for your study). In order to proceed with your submission, you must provide confirmation.

“This work was supported by the National Council for Scientific and Technological Development (CNPq), Brazil.”

“No conflicts of interest are reported for this article.”

5. Please ensure that you refer to Figure 1 in your text as, if accepted, production will need this reference to link the reader to the figure.

Reviewers' comments:

Reviewer's Responses to Questions

**Comments to the Author**

1. Does the manuscript provide a valid rationale for the proposed study, with clearly identified and justified research questions?

Reviewer #1: Yes

Reviewer #2: Partly

2. Is the protocol technically sound and planned in a manner that will lead to a meaningful outcome and allow testing the stated hypotheses?

Reviewer #1: Yes

Reviewer #2: No

3. Is the methodology feasible and described in sufficient detail to allow the work to be replicable?

Reviewer #1: Yes

Reviewer #2: Yes

4. Have the authors described where all data underlying the findings will be made available when the study is complete?

Reviewer #1: Yes

Reviewer #2: No

5. Is the manuscript presented in an intelligible fashion and written in standard English?

Reviewer #1: Yes

Reviewer #2: No

6. Review Comments to the Author

You may also provide optional suggestions and comments to authors that they might find helpful in planning their study.

Reviewer #1: Although the protocol is well-structured and follows robust guidelines for systematic reviews and meta-analyses, some aspects could be improved to enhance the clarity and methodological quality of the work. Below are my suggestions for improvement:

- Publication Bias Assessment: While the use of a random-effects model for the meta-analysis is appropriate, it is unclear how the protocol will address publication bias. It is recommended to include methods for identifying and managing this issue, such as funnel plots or statistical tests like Egger's test.

- Qualitative Data Analysis: The protocol mentions the analysis of categories and emerging themes (qualitative data) but does not clarify under which circumstances these data will be analyzed or presented. Given the quantitative focus of the study, it would be helpful to specify the role and relevance of these qualitative data within the overall scope of the review.

- Consistency in Terminology: Some expressions, such as “binary outcomes,” “continuous outcomes,” and “multiple outcomes,” appear in the text but could be better harmonized to avoid redundancy or confusion. It is recommended to standardize the terminology throughout the protocol to ensure greater clarity.

Reviewer #2: The authors intended to develop a protocol for reviewing studies involving aerobic exercise and depression; however, I believe the study promotes something but missed the opportunity to deliver. There are numerous good examples of systematic reviews, so why should this protocol be followed? Furthermore, there are several inconsistencies, such as ignoring the fact that vigorous-intensity exercise is already recognized as involving anaerobic effort. Claiming that an exercise is aerobic solely based on heart rate and perceived exertion is scientifically imprecise. Measurements of lactate or gas analysis are necessary. Even interval exercise can be aerobic as long as the work-rest ratio does not promote an imbalance in blood lactate levels. Moreover, the grammar and verb tense used in the article suggest to the reader that something was discovered, but in reality, no results are presented.

7. PLOS authors have the option to publish the peer review history of their article (what does this mean? ). If published, this will include your full peer review and any attached files.

**Do you want your identity to be public for this peer review?** For information about this choice, including consent withdrawal, please see our Privacy Policy .

Reviewer #1: No

Reviewer #2: No

---

## [Author Response · Author response to Decision Letter 1]

1 Mar 2025

REBUTTAL LETTER

Manuscript number: PONE-D-24-51714

Title: Effectiveness of aerobic exercise on depression symptoms in adults: A protocol for a systematic review and meta-analysis of randomized clinical trials

Dear Reviewers,

First and foremost, I would like to express my gratitude, on behalf of all the authors of the manuscript, to the esteemed reviewers for raising considerations of utmost importance, which are highly relevant to improving the article.

I would like to emphasize to the distinguished reviewers that the manuscript submitted to PLOS ONE aims to summarize scientific evidence on the effect of aerobic exercise in the treatment of depression in adults. We have made all the requested adjustments, which have been incorporated into the manuscript itself, with the changes highlighted in red font. These modifications are also listed below:

RESPONSE TO REVIEWERS

REVIEWER 1

Reviewer: Although the protocol is well-structured and follows robust guidelines for systematic reviews and meta-analyses, some aspects could be improved to enhance the clarity and methodological quality of the work. Below are my suggestions for improvement:

- Publication Bias Assessment: While the use of a random-effects model for the meta-analysis is appropriate, it is unclear how the protocol will address publication bias. It is recommended to include methods for identifying and managing this issue, such as funnel plots or statistical tests like Egger's test.

authors' response:

For the assessment of publication bias, Egger's statistical test will be implemented, a sophisticated tool used to enhance the consistency of funnel plot analysis [14]. In cases where publication bias is identified, a technique that may be used to correct it is the trim-and-fill method, which adjusts the distribution of studies in an asymmetric funnel plot [15].

- Qualitative Data Analysis: The protocol mentions the analysis of categories and emerging themes (qualitative data) but does not clarify under which circumstances these data will be analyzed or presented. Given the quantitative focus of the study, it would be helpful to specify the role and relevance of these qualitative data within the overall scope of the review.

authors' response:

Quantitative results will include effect size, measures of variability such as confidence intervals, standard deviation, and p-values, as well as other data relevant for interpreting the results.

authors' justification: Most Esteemed Reviewer 1, The authors of this manuscript have conducted a more precise analysis and have collectively decided to exclude the qualitative data from the protocol of this systematic review. This decision is justified by the methodological nature of the included clinical trials. Randomized controlled trials (RCTs) are predominantly quantitative, focusing on objective measures such as clinical outcomes, standardized scales, and statistical analyses. Thus, the expectation of finding relevant qualitative data within this methodological framework is extremely low, which would compromise the feasibility and applicability of the proposed analysis. Additionally, systematic reviews involving clinical trials generally prioritize the assessment of intervention efficacy based on robust quantitative evidence. While the inclusion of qualitative data could be enriching from a phenomenological perspective, it does not align with the primary objectives of this type of review. Incorporating such data would complicate the extraction, analysis, and synthesis of findings in a standardized manner. Therefore, we have opted to remove qualitative data from the protocol to ensure greater methodological rigor and coherence in the approach to the included studies, thereby maintaining the systematic review’s reproducibility within the scientific field.

- Consistency in Terminology: Some expressions, such as “binary outcomes,” “continuous outcomes,” and “multiple outcomes,” appear in the text but could be better harmonized to avoid redundancy or confusion. It is recommended to standardize the terminology throughout the protocol to ensure greater clarity.

authors' response:

The mean and standard deviation (SD) values reflecting the change in depression scores from baseline to post-intervention will be extracted from each study for pooling effects. SD will be calculated using a previously described formula for studies reporting standard error (SE) or 95% confidence intervals (CIs). The effect models (fixed or random) will be used to estimate the treatment effect (aerobic exercise) for depression in adults. The choice of model for analysis will depend on the similarity of the effects of interest among the studies: fixed-effect models will be applied to similar studies, while random-effects models will be used for the others.

The presence of bias in relation to estimates against sample size will be analyzed using a funnel plot. Heterogeneity among studies will be assessed using the I² index, which quantifies inconsistency among study results, and Cochran’s Q test, which evaluates the statistical significance of heterogeneity. Heterogeneity will be interpreted according to I² values following the recommendations of the Cochrane Handbook for Systematic Reviews of Interventions. Statistical significance will be set at P < 0.05. Review Manager software will be used for all analyses, including meta-analysis if feasible [13]. One review author will be responsible for entering data into the software, while a second author will check data accuracy.

authors' justification: Dear Reviewer, we have reorganized the text, enhancing the highlighted aspects related to terminology. In this revised version, coherence between the sections has been improved, making our proposed approach to the data clearer.

REVIEWER 2

- The authors intended to develop a protocol for reviewing studies involving aerobic exercise and depression; however, I believe the study promotes something but missed the opportunity to deliver. There are numerous good examples of systematic reviews, so why should this protocol be followed?

authors' response:

Therefore, this study is justified by the need to understanding of how different intensities and types of aerobic exercise can influence the reduction of depression levels in adults. Various international guidelines, such as those from the United Kingdom and Australia, recommend physical exercise as part of depression treatment [4,5]. However, these guidelines do not provide clear and consistent recommendations regarding the dose or modality of exercise [6]. In turn, the American Psychiatric Association recommends any dose of aerobic exercise or strength training [7], while the Australian guidelines already suggest a combination of vigorous aerobic and strength exercises at least twice a week [5].

Supposedly, a network meta-analysis has shown that dose and modality influence cognition [8]. In this regard, it is essential to consider the need for reviews like this one, given the numerous designs of meta-analyses and systematic reviews that, despite their relevance, have presented methodological limitations preventing fundamental conclusions. Under these conditions, considering the isolated analysis of aerobic exercises may be a promising alternative, allowing for a more robust assessment of the accuracy of their effects, as there is a well-documented relationship in the literature regarding this type of exercise.

A recently conducted systematic review investigating the effect of physical exercise on depression revealed that dance exercises combined with selective serotonin reuptake inhibitors (SSRIs), walking, or running were the treatments most likely to be effective in treating depression. Furthermore, the results indicate that the effect of exercise is proportional to its intensity, with vigorous activities showing better responses. However, the findings did not allow for the precise establishment of a dose-response relationship due to variability in the study protocols analyzed, particularly regarding frequency, duration, and weekly volume of activity [6]. In this regard, conducting the proposed review with aerobic exercises may be a viable option for refining the results.

authors' justification: Dear Reviewer, the authors of this manuscript have revised the rationale for conducting this systematic review based on the existing literature. The main objective was to overcome the limitations identified in previous reviews, enhancing the study's approach and theoretical foundation.

- Furthermore, there are several inconsistencies, such as ignoring the fact that vigorous-intensity exercise is already recognized as involving anaerobic effort. Claiming that an exercise is aerobic solely based on heart rate and perceived exertion is scientifically imprecise. Measurements of lactate or gas analysis are necessary. Even interval exercise can be aerobic as long as the work-rest ratio does not promote an imbalance in blood lactate levels.

authors' response:

In this context, physical exercise emerges as a promising intervention for improving mental health. The focus of this review will be the study of exercises with an aerobic predominance, commonly known as 'aerobic exercise,' which are characterized by the primary use of the oxidative pathway for energy production. This type of exercise involves long-duration activities with varying intensity levels, ranging from low (walking), moderate (light jogging, cycling), to high (continuous running near VO₂max, competitive cycling, endurance swimming). The energy demand in these efforts is primarily sustained by mitochondrial metabolism [2].

However, the analysis of these exercises should not be based solely on perceived effort or heart rate, as all energy pathways operate simultaneously, varying according to the duration and intensity of the exercise [2]. Thus, efforts are categorized as follows: explosive efforts (up to 6 seconds), dominated by the phosphagen system (ATP-CP); high-intensity efforts (6 seconds to 1 minute), where anaerobic glycolysis predominates but with an increasing aerobic contribution; and prolonged endurance efforts (above 1–2 minutes), where oxidative phosphorylation plays a primary role, although anaerobic glycolysis still contributes during intensity transition phases [2].

authors' justification: After a detailed analysis of the text regarding the definition of aerobic exercises, the authors recognized the need to refine the presented conceptualization. In fact, the observation made by the reviewer was highly relevant. Considering the inconsistency pointed out, we revised the passage to emphasize the dynamic interaction between energy systems, avoiding the classification of an exercise as aerobic solely based on heart rate or perceived exertion.

- Moreover, the grammar and verb tense used in the article suggest to the reader that something was discovered, but in reality, no results are presented.

authors' justification:

The request for grammatical revision and verb tense adjustment in the manuscript has been addressed, with the changes highlighted in the discussion. We hope that the modifications made are aligned with the requested adjustments. We also appreciate all the considerations and suggestions provided, as they were of great importance for the improvement of this review and contributed significantly to the field of scientific knowledge.

The reference to Figure 1 was made in the text on page 6 of the manuscript and highlighted in red font.

---

## [Decision Letter · Decision Letter 1]

25 Mar 2025

PONE-D-24-51714R1Effectiveness of aerobic physical exercise on depression symptoms in adults: A protocol for a systematic review and meta-analysis of randomized clinical trialsPLOS ONE

Dear Dr. Souza,

Thank you for submitting your manuscript to PLOS ONE. After careful consideration, we feel that it has merit but does not fully meet PLOS ONE’s publication criteria as it currently stands. Therefore, we invite you to submit a revised version of the manuscript that addresses the points raised during the review process.

**ACADEMIC EDITOR: Dear Author, please revise your manuscript based on the comments provided by the reviewer/s.**

We look forward to receiving your revised manuscript.

Kind regards,

Zulkarnain Jaafar

Academic Editor

PLOS ONE

Journal Requirements:

Reviewers' comments:

Reviewer's Responses to Questions

**Comments to the Author**

1. Does the manuscript provide a valid rationale for the proposed study, with clearly identified and justified research questions?

Reviewer #2: No

Reviewer #3: Yes

2. Is the protocol technically sound and planned in a manner that will lead to a meaningful outcome and allow testing the stated hypotheses?

Reviewer #2: No

Reviewer #3: Yes

3. Is the methodology feasible and described in sufficient detail to allow the work to be replicable?

Reviewer #2: Yes

Reviewer #3: Yes

4. Have the authors described where all data underlying the findings will be made available when the study is complete?

Reviewer #2: No

Reviewer #3: Yes

5. Is the manuscript presented in an intelligible fashion and written in standard English?

Reviewer #2: Yes

Reviewer #3: Yes

6. Review Comments to the Author

You may also provide optional suggestions and comments to authors that they might find helpful in planning their study.

Reviewer #2: It seems like you're pointing out that the research was more of a plan or outline rather than an actual study, without concrete results.

Reviewer #3: Dear authors

The article presents scientific merit and contributes to the advancement of scientific knowledge about an efficient roadmap (a strategy) for conducting systematic review studies and meta-analysis of randomized clinical trials. The authors describe the methodological procedures for conducting scientific studies of this nature using different types of aerobic physical exercise and their outcomes related to symptoms of depression in adults. The protocol outlined in the article can be applied in conducting this study design to minimize the risk of methodological errors.

However, I suggest some adjustments to improve the article.

I suggest changing the title of the article to: Effectiveness of aerobic physical exercise on depression symptoms in adults: A protocol for developing a systematic review and meta-analysis of randomized clinical trials

Introduction:

The authors must make it clear that the aim of this study was to develop a protocol of methodological procedures to conduct systematic review studies with meta-analysis of randomized clinical trials on different types of aerobic physical exercises and their outcomes related to symptoms of depression in adults.

Methods:

Authors must include the reference to the PICOS strategy (Methley AM, Campbell S, Chew-Graham C, McNally R, Cheraghi-Sohi S. PICO, PICOS and SPIDER: a comparison study of specificity and sensitivity in three search tools for qualitative systematic reviews. BMC Health Serv Res. 2014;14:579. doi:10.1186/s12913-014-0579-0). In this section, there are repeated passages about the PICOS strategy. I suggest leaving only one citation in the text.

Authors can indicate the TESTEX tool (Smart NA, Waldron M, Ismail H, et al. Validation of a new tool for the assessment of study quality and reporting in exercise training studies: TESTEX. Int J Evid Based Healthcare. 2015;13(1):9–18. doi:10. 1097/XEB.0000000000000020), because it is more suitable for assessing the methodological quality of studies involving physical exercise. While RoB2 will be used to assess the risk of bias.

Authors must indicate the I2 values to use fixed or random effects in the decision-making process in the meta-analysis (Forest-Plot).

Authors must indicate the classification and application of the categories in the GRADE tool.

Authors must indicate when a funnel plot should be used (minimum of 10 articles).

Authors must indicate the main points of analysis and interpretation of a forest plot.

Discussion:

In the discussion section, authors must highlight in depth the main factors that can generate high heterogeneity (I2) in the meta-analysis and how this can be considered as one of the limitations of a systematic review study with meta-analysis.

Sincerely

7. PLOS authors have the option to publish the peer review history of their article (what does this mean? ). If published, this will include your full peer review and any attached files.

**Do you want your identity to be public for this peer review?** For information about this choice, including consent withdrawal, please see our Privacy Policy .

Reviewer #2: No

Reviewer #3: No

---

## [Author Response · Author response to Decision Letter 2]

31 Mar 2025

Response to Reviewers

Manuscript number: PONE-D-24-51714

Title: Effectiveness of aerobic physical exercise on depression symptoms in adults: A protocol for developing a systematic review and meta-analysis of randomized clinical trials

Dear Reviewers,

First and foremost, I would like to express my gratitude, on behalf of all the authors of the manuscript, to the esteemed reviewers for raising extremely important considerations that are highly relevant for improving the article.

It is important to note to the distinguished reviewers that the manuscript submitted to PLOS ONE is a robust protocol for conducting a systematic review, aimed at gathering scientific evidence on the effect of aerobic exercise in the treatment of depression in adults.

We have made all the requested adjustments, which have been incorporated into the manuscript itself, with the changes highlighted in red font.

These modifications are also listed below:

Reviewer #2 Authors' response:

Reviewer: It seems like you're pointing out that the research was more of a plan or outline rather than an actual study, without concrete results.

We appreciate your contribution and agree with your position. Please review the changes that were made to some parts of the abstract and the introduction.

In the abstract/objective: “This article presents a protocol for conducting a systematic review with meta-analysis, aimed at investigating the effects of aerobic exercise on reducing depression symptoms in adults.”

In the abstract/methods: “Data extraction will be performed using specific forms, while the methodological quality of studies on physical exercise will be assessed with the TESTEX tool, and the risk of bias will be evaluated using the Cochrane RoB 2.0 method”.

In the abstract/conclusion: “It is expected that the systematic review following this protocol will identify the effective dose-response to reduce depression levels and provide an understanding of the mechanisms through which aerobic exercise influences depression”.

Some parts of the introduction:

“Although the benefits of aerobic exercise in improving depressive symptoms are well documented, the existing literature shows high variability in terms of modality, intensity, frequency, and duration of exercise. Furthermore, the underlying mechanisms of these effects are still not fully understood [3].“

“Recent reviews, including network meta-analyses, suggest that both the dose and the modality of exercise may influence clinical outcomes [8]. However, many existing systematic reviews and meta-analyses have methodological limitations that prevent definitive conclusions. In light of this, the isolated investigation of aerobic exercise using a robust and clearly defined methodology may contribute to a more accurate understanding of its effectiveness on depressive symptoms.”

“In this regard, the development of this protocol focused exclusively on aerobic exercises represents a viable strategy to guide a future systematic review and refine the results obtained in the scientific literature.

This protocol proposal aims to guide a future systematic review that will seek to answer the following question: what are the effects of aerobic exercise on reducing depression levels in adults? With this protocol, we intend to provide researchers and healthcare professionals with a rigorous and transparent methodological foundation for conducting a systematic review and meta-analysis of randomized clinical trials on the topic.”

Reviewer #3

The article presents scientific merit and contributes to the advancement of scientific knowledge about an efficient roadmap (a strategy) for conducting systematic review studies and meta-analysis of randomized clinical trials. The authors describe the methodological procedures for conducting scientific studies of this nature using different types of aerobic physical exercise and their outcomes related to symptoms of depression in adults. The protocol outlined in the article can be applied in conducting this study design to minimize the risk of methodological errors.

However, I suggest some adjustments to improve the article.

- I suggest changing the title of the article to: Effectiveness of aerobic physical exercise on depression symptoms in adults: A protocol for developing a systematic review and meta-analysis of randomized clinical trials We appreciate your input and concur with your position. Kindly review the alterations that have been made.

“Effectiveness of aerobic physical exercise on depression symptoms in adults: A protocol for developing a systematic review and meta-analysis of randomized clinical trials”

Introduction:

- The authors must make it clear that the aim of this study was to develop a protocol of methodological procedures to conduct systematic review studies with meta-analysis of randomized clinical trials on different types of aerobic physical exercises and their outcomes related to symptoms of depression in adults. We appreciate your input and concur with your position. Kindly review the alterations that have been made.

“This protocol proposal aims to guide a future systematic review that will seek to answer the following question: what are the effects of aerobic exercise on reducing depression levels in adults? With this protocol, we intend to provide researchers and healthcare professionals with a rigorous and transparent methodological foundation for conducting a systematic review and meta-analysis of randomized clinical trials on the topic.”

-Methods:

Authors must include the reference to the PICOS strategy (Methley AM, Campbell S, Chew-Graham C, McNally R, Cheraghi-Sohi S. PICO, PICOS and SPIDER: a comparison study of specificity and sensitivity in three search tools for qualitative systematic reviews. BMC Health Serv Res. 2014;14:579. doi:10.1186/s12913-014-0579-0).

- In this section, there are repeated passages about the PICOS strategy. I suggest leaving only one citation in the text. We appreciate your input and concur with your position. Kindly review the alterations that have been made.

“Thus, the clinical PICOS question, widely recommended to guide systematic reviews [10], will be formulated as follows: Population (adults); Intervention (aerobic exercise); Comparator (other types of exercise/placebo/control); Outcomes (improvement of depressive symptoms); Study types (randomized controlled trials – RCTs).”

“Eligibility criteria

We will follow these criterias: Participants will include adults (>18 years), excluding individuals with chronic degenerative diseases or severe psychiatric disorders. The intervention will focus on aerobic exercise (e.g., walking, running), with comparators including other interventions or control groups. The primary outcome will be depressive symptoms, while training volume and frequency will be considered secondary outcomes. Only studies utilizing validated depression assessment scales will be included, such as the Beck Depression Inventory (BDI), Hamilton Depression Rating Scale (HAM-D), Patient Health Questionnaire (PHQ-9), Center for Epidemiologic Studies Depression Scale (CES-D), Montgomery-Åsberg Depression Rating Scale (MADRS), Zung Self-Rating Depression Scale (Zung SDS), Geriatric Depression Scale (GDS), and Hospital Anxiety and Depression Scale (HADS), among others. Eligible study types will be randomized clinical trials (RCTs), no restriction of language, while ongoing clinical trials with preliminary results will be excluded. Additionally, studies involving populations with non-communicable chronic diseases (e.g., hypertension, diabetes), case reports, preprints, narrative and systematic reviews, and observational studies will not be considered for inclusion.”

- Authors can indicate the TESTEX tool (Smart NA, Waldron M, Ismail H, et al. Validation of a new tool for the assessment of study quality and reporting in exercise training studies: TESTEX. Int J Evid Based Healthcare. 2015;13(1):9–18. doi:10. 1097/XEB.0000000000000020), because it is more suitable for assessing the methodological quality of studies involving physical exercise. While RoB2 will be used to assess the risk of bias.

We appreciate your input and concur with your position. Kindly review the alterations that have been made.

“The methodological quality assessment of the included studies will be carried out by two independent reviewers (L.N.S; M.I.K) using the TESTEX tool (Tool for the assEssment of Study qualiTy and reporting in EXercise), which was specifically developed for clinical trials involving physical exercise interventions. This scale, validated by Smart et al. [12], includes criteria addressing both methodological quality and reporting quality of the studies, with a total possible score of 15 points—5 for study quality and 10 for reporting quality. TESTEX stands out for addressing relevant and specific aspects of exercise interventions, such as adherence to the protocol, exercise intensity, and volume, making it more appropriate than other tools for use in exercise-related reviews.

In addition, the Cochrane Risk of Bias 2.0 tool (RoB 2.0) will be used to assess the risk of bias in the included studies [13]. This tool evaluates the following domains: the randomization process, deviations from the intended intervention, missing outcome data, measurement of outcomes, selection of the reported results, and the overall judgment of the risk of bias.”

- Authors must indicate the I2 values to use fixed or random effects in the decision-making process in the meta-analysis (Forest-Plot).

We appreciate your input and concur with your position. Kindly review the alterations that have been made.

“The meta-analysis will be conducted using fixed-effect or random-effects models, depending on the heterogeneity among the included studies. The degree of heterogeneity will be assessed using the I² statistic. A fixed-effect model will be applied when heterogeneity is low (I² ≤ 50%), while a random-effects model will be used when heterogeneity is moderate or high (I² > 50%) [15].”

- Authors must indicate the classification and application of the categories in the GRADE tool. We appreciate your input and concur with your position. Kindly review the alterations that have been made.

“In this protocol, the GRADE approach will be applied to assess the certainty of the evidence for each outcome included in the systematic review and meta-analysis. The evaluation will be based on the following domains: risk of bias, inconsistency of results, indirectness of evidence, imprecision of estimates, and potential publication bias. Two reviewers will independently assess each domain, and disagreements will be resolved by consensus or by a third reviewer. The results will be summarized in a GRADE evidence profile and presented in a Summary of Findings table. GRADE categorizes both the level of evidence and the strength of recommendations into four categories: high, moderate, low, and very low [14].”

- Authors must indicate when a funnel plot should be used (minimum of 10 articles). We appreciate your input and concur with your position. Kindly review the alterations that have been made.

“The presence of bias in relation to estimates against sample size will be analyzed using a funnel plot, which will be generated only if at least 10 studies are included in the meta-analysis, as recommended to ensure reliability of asymmetry assessments [15].”

- Authors must indicate the main points of analysis and interpretation of a forest plot. We appreciate your input and concur with your position. Kindly review the alterations that have been made.

“We will analyze a forest plot by considering the effect size and 95% confidence intervals (CIs), heterogeneity (I²), study weight, and the pooled effect size. All of these points should be interpreted in relation to the null value, and potential publication bias should be evaluated using a funnel plot or Egger’s test.”

Discussion:

In the discussion section, authors must highlight in depth the main factors that can generate high heterogeneity (I2) in the meta-analysis and how this can be considered as one of the limitations of a systematic review study with meta-analysis. We appreciate your input and concur with your position. Kindly review the alterations that have been made.

“The systematic review studies conducted so far, although following rigorous methodologies and providing relevant contributions to the scientific field, present some methodological limitations. Anticipated limitations and challenges include the following issues: the use of small sample sizes, lack of clarity regarding the specific mechanisms through which exercise contributes to the improvement of depression, and variations in the intensity protocols adopted. In addition, one of the main methodological challenges expected in the conduction of this review is the high heterogeneity among the studies to be included. Differences may arise in exercise modalities, levels of depressive symptoms, different intensities, frequency and duration of exercise, presence of other comorbidities, supervised versus unsupervised exercise, individual versus group exercise, and gender differences, as men and women may respond differently. However, this review aims precisely to highlight these issues and deepen the understanding of their implications.”

---

## [Decision Letter · Decision Letter 2]

23 Apr 2025

PONE-D-24-51714R2Effectiveness of aerobic physical exercise on depression symptoms in adults: A protocol for developing a systematic review and meta-analysis of randomized clinical trialsPLOS ONE

Dear Dr. Souza,

Thank you for submitting your manuscript to PLOS ONE. After careful consideration, we feel that it has merit but does not fully meet PLOS ONE’s publication criteria as it currently stands. Therefore, we invite you to submit a revised version of the manuscript that addresses the points raised during the review process.

We look forward to receiving your revised manuscript.

Kind regards,

Zulkarnain Jaafar

Academic Editor

PLOS ONE

Journal Requirements:

Additional Editor Comments:

**Dear Author, please make revisions based on the reviewer/s' comments and adjust your manuscript accordingly.**

Reviewers' comments:

Reviewer's Responses to Questions

**Comments to the Author**

1. Does the manuscript provide a valid rationale for the proposed study, with clearly identified and justified research questions?

Reviewer #3: Yes

Reviewer #4: Partly

2. Is the protocol technically sound and planned in a manner that will lead to a meaningful outcome and allow testing the stated hypotheses?

Reviewer #3: Yes

Reviewer #4: Yes

3. Is the methodology feasible and described in sufficient detail to allow the work to be replicable?

Reviewer #3: Yes

Reviewer #4: Yes

4. Have the authors described where all data underlying the findings will be made available when the study is complete?

Reviewer #3: Yes

Reviewer #4: Yes

5. Is the manuscript presented in an intelligible fashion and written in standard English?

Reviewer #3: Yes

Reviewer #4: Yes

6. Review Comments to the Author

You may also provide optional suggestions and comments to authors that they might find helpful in planning their study.

Reviewer #3: Dear authors

The article presents scientific merit and contributes to the advancement of scientific knowledge about an efficient roadmap (a strategy) for conducting systematic review studies and meta-analysis of randomized clinical trials.

The authors worked well on the requested changes. All my questions were answered satisfactorily.

I now believe that the protocol will be able to guide researchers for developing a systematic review with meta-analysis with fewer chances of errors. These procedures reported in the article may increase the level of assertiveness and scientificity of the information obtained in studies with this type of characteristics. The protocol outlined in the article can be applied in conducting this study design to minimize the risk of methodological errors.

Sincerely,

Reviewer #4: General Comments:

Many thanks for the invitation to review this manuscript. In this paper the authors present a protocol for a systematic review and meta-analysis of randomized clinical trials on the effectiveness of aerobic physical exercise on depression symptoms in adults. The study has good elements of originality. Below are few suggested minor comments to address.

Introduction

Please incorporate literature on the impact of depression and draw these together to urgency for effective strategies

Methods

Participants: It is not clear whether participants would need to have depression or not. Please indicate this.

7. PLOS authors have the option to publish the peer review history of their article (what does this mean? ). If published, this will include your full peer review and any attached files.

**Do you want your identity to be public for this peer review?** For information about this choice, including consent withdrawal, please see our Privacy Policy .

Reviewer #3: No

Reviewer #4: No

---

## [Author Response · Author response to Decision Letter 3]

24 Apr 2025

Response to Reviewers

Manuscript number: PONE-D-24-51714

Title: Effectiveness of aerobic physical exercise on depression symptoms in adults: A protocol for developing a systematic review and meta-analysis of randomized clinical trials

Dear Reviewers,

First and foremost, I would like to express my gratitude, on behalf of all the authors of the manuscript, to the esteemed reviewers for raising extremely important considerations that are highly relevant for improving the article.

It is important to note to the distinguished reviewers that the manuscript submitted to PLOS ONE is a robust protocol for conducting a systematic review, aimed at gathering scientific evidence on the effect of aerobic exercise in the treatment of depression in adults.

We have made all the requested adjustments, which have been incorporated into the manuscript itself, with the changes highlighted in red font.

These modifications are also listed below:

Reviewer #4 Authors' response:

Reviewer: Many thanks for the invitation to review this manuscript. In this paper the authors present a protocol for a systematic review and meta-analysis of randomized clinical trials on the effectiveness of aerobic physical exercise on depression symptoms in adults. The study has good elements of originality. Below are few suggested minor comments to address.

Introduction

- Please incorporate literature on the impact of depression and draw these together to urgency for effective strategies

Methods

- Participants: It is not clear whether participants would need to have depression or not. Please indicate this.

We appreciate your input and concur with your position. Kindly review the alterations that have been made.

“Since it is a condition often aggravated by other diseases, such as substance use disorders, diabetes, and heart diseases, which not only increase the risk of developing depression but are also affected by it, establishing a cycle that further compromises the health and quality of life of those affected [2].

In general, mental health problems, neurological disorders, and substance use disorders account for 13% of the global burden of disease, with depression alone responsible for 4.3% of this total, women are the most affected [3]. In 2019, there were 290.2 million cases of depression worldwide, an increase of 59.28% compared to 1990 [3].

Projections based on the Bayesian age-period-cohort (BAPC) model indicate that, by 2030, global age-standardized rates are expected to remain stable. However, the absolute number of depression cases is likely to increase, potentially reaching 108.9 million among men and 164.9 million among women [4].

The research conducted by Liu et al. [5] reinforces these projections by indicating stability in age-specific incidence rates between 1990 and 2017, along with an increase in the total number of global depression cases, reaching 258 million in 2017, a growth of 49.86% compared to 1990. Regions with low Socio-Demographic Index (SDI) face the greatest impacts, mainly due to the lack of investment in mental health. In these areas, depression often remains undiagnosed or inadequately treated, resulting in higher morbidity and disability.

In the United States, the annual prevalence of depression among adults reached 8.9 million people undergoing pharmacological treatment, with approximately 30.9% (2.8 million) being treatment-resistant [6].

Treatment-resistant depression is defined as the failure of at least two adequate trials with different antidepressants, over a minimum period of four to six weeks, at the maximum recommended dose [7,8].

Given the multifaceted nature of the disorder, significant gaps are still observed in the studies, attributed in part to the diversity of symptoms and the limitations of current technologies to investigate the human brain in real time, both at the circuit and synapse levels [9]. The condition can be triggered by biological changes [10], as well as by psychosocial influences [11,12].

In view of the aforementioned issues, researchers and scholars in the field have been seeking effective strategies capable of responding, in an individualized manner, to the complexities of depressive disorder, considering its multifactorial nature. In this context, physical exercise has emerged as a promising intervention for mental health, especially among individuals with antidepressant-resistant depression. Furthermore, evidence indicates that even in these cases, physical exercise, when used as an adjunct to pharmacological treatment, demonstrates significant potential for reducing depressive symptoms, including non-remissive cases, as well as improving quality of life, sleep, and vitality [13]”.

- In the Research Question section, we added 'adults with depression' to the population.

“Thus, the clinical PICOS question, widely recommended to guide systematic reviews [22], will be formulated as follows: Population (adults with depression); Intervention (aerobic exercise); Comparator (other types of exercise/placebo/control); Outcomes (improvement of depressive symptoms); Study types (randomized controlled trials – RCTs).”

- In the Eligibility Criteria section, we also added the indication at the beginning of the paragraph.

“Participants will include adults (>18 years) diagnosed with depression, excluding individuals with chronic degenerative diseases or other associated severe psychiatric disorders.”

---

## [Editor Report · Decision Letter 3]

28 Apr 2025

Effectiveness of aerobic physical exercise on depression symptoms in adults: A protocol for developing a systematic review and meta-analysis of randomized clinical trials

PONE-D-24-51714R3

Dear Dr. Souza,

We’re pleased to inform you that your manuscript has been judged scientifically suitable for publication and will be formally accepted for publication once it meets all outstanding technical requirements.

Kind regards,

Zulkarnain Jaafar

Academic Editor

PLOS ONE
---

## [Editor Report · Acceptance letter]

PONE-D-24-51714R3

PLOS ONE

Dear Dr. Souza,

I'm pleased to inform you that your manuscript has been deemed suitable for publication in PLOS ONE. Congratulations! Your manuscript is now being handed over to our production team.

Kind regards,

on behalf of

Dr. Zulkarnain Jaafar

Academic Editor

PLOS ONE